# Prevalence and Genetic Characteristics of Avian Chlamydia in Birds in Guangxi, Southwestern China

**DOI:** 10.3390/microorganisms13092220

**Published:** 2025-09-22

**Authors:** Jian-Ming Long, Hai-Tao Zhong, Ya-Yu Deng, Jun-Wei Yang, Mei-Chi Chen, Yan-Jiao Liang, Ke-Wei Chen, Jing-Ting Yang, Tian-Chao Wei, Ping Wei, Jian-Ni Huang

**Affiliations:** 1College of Animal Science and Technology, Guangxi University, Nanning 530004, China; longjm2025@163.com (J.-M.L.); zhonght2025@163.com (H.-T.Z.); 15077556993@163.com (Y.-Y.D.); 17655993115@163.com (J.-W.Y.); 17381092659@163.com (M.-C.C.); liangyanjiao1219@163.com (Y.-J.L.); kewei286513@163.com (K.-W.C.); yjt15650122411@163.com (J.-T.Y.); tcwei88@126.com (T.-C.W.); 2Guangxi Key Laboratory of Animal Breeding, Disease Control and Prevention, Nanning 530004, China; 3Guangxi Zhuang Autonomous Region Engineering Research Center of Veterinary Biologics, Nanning 530004, China

**Keywords:** avian chlamydiosis, prevalence, genetic characteristics, bird

## Abstract

Avian chlamydiosis, primarily caused by *Chlamydia psittaci* (*C. psittaci*), poses significant threats to poultry and avian trade. Emerging species such as *Chlamydia gallinacea* (*C. gallinacea*), *Chlamydia avium* (*C. avium*), and *Chlamydia ibidis* (*C. ibidis*) have recently been detected in birds. However, the prevalence and genetic diversity of avian chlamydia in birds within Guangxi remain unknown. In this study, 1744 samples collected from apparently healthy birds were screened, revealing an overall positivity rate of 28.20% (95% CI, 27.58–28.90%, 492/1744) for avian chlamydia. Among poultry, pigeons had the highest positivity rate at 62.30% (95% CI, 55.37–68.69%, 152/244), followed by chickens at 25.05% (95% CI, 21.25–29.23%, 128/511), geese at 18.12% (95% CI, 12.93–24.82%, 29/160), and ducks at 14.14% (95% CI, 11.57–17.26%, 82/580). Additionally, pet and wild birds exhibited positivity rates of 40.35% (95% CI, 34.20–46.83%, 92/228) and 42.86% (95% CI, 24.52–61.83%, 9/21), respectively. Phylogenetic analysis based on the outer-membrane protein A gene indicated that chicken samples belonged to genotypes B of *C. psittaci* and *C. gallinacea*. In ducks, genotypes A and B of *C. psittaci* and *C. gallinacea* were identified, representing the first documented occurrence of *C. psittaci* genotypes B and *C. gallinacea* in ducks in China. The nucleotide sequences from goose samples were initially clustered into genotype A group, while those from pigeons were clustered within genotype B. Furthermore, positive samples from pet birds were classified into genotypes A and B, as well as the *C. gallinacea* group. Similarly, samples from wild birds were classified into genotypes A and B. These findings suggest that diverse avian chlamydia genotypes are circulating among bird populations in Guangxi, with an expanding host range indicating potential cross-species transmission. Moreover, certain strains derived from waterfowl were found to cluster with those linked to recent psittacosis outbreaks, highlighting the zoonotic potential of avian chlamydia. Therefore, sustained surveillance for avian chlamydia in bird populations and monitoring its genetic evolutionary characteristics are essential to decrease public health risks.

## 1. Introduction

Avian chlamydiosis is an infectious disease caused by *Chlamydia* species in birds. The primary causative agent of this disease is the obligate intracellular Gram-negative bacterium *Chlamydia psittaci* (*C. psittaci*). The clinical signs, including respiratory, digestive, systemic, and occasionally fatal manifestations, occur in most infected birds [1,2,3]. The widespread distribution of this infection in poultry results in economic losses for birds raised commercially for meat and egg production, as well as an increased risk of zoonotic transmission to humans [4].

Globally, *C. psittaci* is a well-known *Chlamydia* species that infects more than 500 avian species belonging to 30 orders, including poultry, pet birds, and wild birds [2,5,6]. Moreover, various mammals, such as cattle, sheep, and horses, have been reported as reservoirs of this pathogen [3,7]. *C. psittaci* is also regarded as a potential zoonotic agent, causing psittacosis, a non-specific influenza-like illness or atypical pneumonia. Human cases are often associated with close contact with infected birds in the poultry industry or occupational exposure to birds, such as pet shop employees, veterinarians, or poultry plant workers [8,9,10]. *C. psittaci* is divided into 15 genotypes based on the outer membrane protein A (*OmpA*) gene, with 13 genotypes originating from birds (A–F, E/B, 1V, 6N, Mat116, R54, YP84, and CPX0308) and two genotypes from mammals (M56 and WC). Distinct genotypes have been identified from a range of host species: genotypes A and F are predominantly found in psittacine birds, genotype B in pigeons, genotype C in ducks and geese, genotype D in turkeys, and genotype E in a wide variety of hosts, including pigeons, turkeys, ducks, ratites, and humans. Genotype E/B is mainly detected in ducks, while other genotypes (1V, 6N, Mat116, R54, YP84, and CPX0308) have been identified in birds [11,12,13,14].

In the past decade, new agents of avian chlamydia, including *Chlamydia gallinacea* (*C. gallinacea*), *Chlamydia avium* (*C*. *avium*), *Chlamydia ibidis* (*C*. *ibidis*), and *Chlamydia buteonis*, have been described [15,16,17,18]. Since 2009, *C. gallinacea* has been frequently detected in chickens across many countries, with positive rates ranging from 12.4% to 81.2% [19,20,21,22,23,24,25]. However, its capacity to cause illness seems limited, as studies have primarily shown decreased body weight gain in broiler chickens [20]. Recently, *C. gallinacea* infections have also been identified in other poultry and wild birds [26,27]. Additionally, whole blood samples, milk, feces, and vaginal swabs collected from healthy dairy and beef cattle in China have tested positive for *C. gallinacea*, suggesting the potential for *C. gallinacea* transmission from birds to ruminants. *C. avium* was first reported in urban pigeons in Germany and France, with positive rates of 3% (4/128) and 8% (10/125), respectively [28,29]. Although sporadic cases have been reported in psittacine birds, *C. avium* is mainly detected in pigeons [30].

Human psittacosis cases have been widely reported across the globe, including in China. Sporadic and spillover infections are often linked to occupational exposure, particularly in the poultry industry, as well as environmental contact with infected birds [31,32,33]. The zoonotic potential of *C. gallinacea* and *C. avium* remains poorly understood due to the lack of species-specific serological diagnostic tools. However, cases of atypical pneumonia in slaughterhouse workers have been associated with exposure to chickens infected with *C. gallinacea* [17]. This underscores the necessity of surveillance and monitoring the epidemiology of avian chlamydia in both avian and human populations.

Guangxi, situated in southwestern China, spans the subtropical and tropical regions. Bordering Southeast Asian nations, Guangxi serves as a critical area for migratory bird breeding and stopover habitats, as well as a significant hub for avian trade activities. Notably, Guangxi ranks fourth nationally in poultry turnover and meat production in China, with yellow-feathered broiler ranking second nationwide. However, available epidemiological data on avian chlamydia infection in birds in China are limited. This study aimed to investigate the prevalence and genetic diversity of avian chlamydia in birds in Guangxi province, located in southwestern China. Additionally, considering the zoonotic potential of these species, we also assessed the potential risk of avian chlamydia for bird-to-human transmission.

## 2. Materials and Methods

### 2.1. Sample Collection

#### 2.1.1. Sampling of Poultry Farms

Between May 2022 and December 2023, authorised veterinarians collected combined cloacal and oropharyngeal swab samples from randomly selected, apparently healthy poultry, adhering to standard procedures. The study focused on large-scale intensive poultry farming companies, collecting approximately 25 combined swabs from each farm within the selected companies. To ensure a defined physical distance during sample collection, farms within the same company were spatially separated. Sampling was conducted across nine regions of Guangxi (Figure 1). The combined swabs (*n* = 511) were collected from chicken flocks across eight farming companies, comprising twelve breeder chicken farms (*n* = 296), five broiler chicken farms (*n* = 100), and five layer chicken farms (*n* = 115). Additionally, 580 combined swabs were obtained from five duck farming companies, including 75 swabs from three breeder mallard duck farms, 225 swabs from nine breeder cherry valley duck farms, 180 swabs from eight broiler cherry valley duck farms, and 100 swabs from four layer duck farms. For geese, 160 combined swabs were collected from seven meat goose farms belonging to two farming companies. Moreover, a total of 244 combined swabs were obtained from meat pigeons across ten farms operated by three farming companies. All combined cloacal and oropharyngeal swabs were preserved in 1 mL of DNA/RNA stabilization reagent and stored at –80 °C until DNA extraction was performed for molecular analysis. The contaminated samples by leakage from collection tubes during transportation were discarded and considered unsuitable for further testing. Detailed information regarding poultry samples is provided in Appendix A.

#### 2.1.2. Sampling of Pet Birds

The zoo implemented an annual vaccination program for pet birds exhibited to the public during the spring and autumn seasons. To investigate the prevalence of avian chlamydia among these birds, combined cloacal and oropharyngeal swabs (*n* = 80) were collected from nine species prior to inoculation in October 2022 and March 2023. The pet market, located in PingXiang (Guangxi, Chongzuo), a border city adjacent to Vietnam, serves as a central hub for the regular trade of pet birds and exhibits a diverse range of captive bird species. Considering the ongoing transactions involving pet birds, a total of 140 cloacal swabs were obtained from six species in December 2022. These included seven samples from blue-breasted quail, fifty-five from quail, nine from zebra finch, twelve from white Java sparrow, thirty-nine from budgerigar, and eighteen from antipodes green parakeet. All combined cloacal and oropharyngeal swabs were preserved in 1 mL of DNA/RNA stabilization reagent. Samples that showed leakage during transportation were discarded and deemed unsuitable for further analysis. The detailed information regarding pet bird samples was present in Appendix A.

#### 2.1.3. Sampling of Wild Birds

Beihai, a coastal city in Guangxi, is situated along the East Asian–Australasian Flyway and serves as a habitat for thousands of migratory and resident waterbirds annually within its mangrove ecosystems. A local pet hospital collaborates with wildlife rehabilitation centers to rescue injured wild birds. During the rescue operations, veterinarians from the pet hospital collected the combined cloacal and oropharyngeal swabs, which were then stored in 1 mL of DNA/RNA stabilization reagent at −80 °C. In this study, twenty-one swabs were obtained from ten species of wild birds, including two cinereous vultures, one savannah nightjar, six red turtle doves, three oriental turtle doves, three spotted doves, two besras, one peregrine falcon, one egret, one great cormorant, and one eastern grass-owl. The detailed information of wild bird samples was seen in Appendix A.

### 2.2. DNA Extraction and Avian Chlamydia Detection

Given the intermittent shedding of chlamydia, the cloacal and oropharyngeal swab samples were pooled as a composite sample for detection. Total DNA from each combined swab was isolated using a bacterial genome DNA extraction kit (Tiangen Biotech, Beijing, China) following the manufacturer’s guidelines. All harvested DNA was stored at –80 °C for further study. To evaluate the prevalence of avian chlamydia in birds in Guangxi, harvested DNA was analyzed using a real-time polymerase chain reaction (PCR) assay according to previous studies [34,35]. Specific primers, PCR-F1 (5′-TCCTTACAAGCCTTGCCTGTA-3′) and PCR-R1 (5′-CACGATCGAAAACATAATCTCCG-3′), were designed based on highly conserved regions of the *ompA* gene from representative strains: *C. psittaci* (GenBank No. X56980.1), *C. gallinacea* (GenBank No. CP019792.1), *C. avium* (GenBank No. CP006571.1), and *C. ibidis* (GenBank No. KE360208.1). The amplifications were carried out in a total reaction volume of 20 μL, consisting of 10 μL of 2× SYBR Green qPCR Master Mix (Bimake, Shanghai, China), 1 μL of DNA, 1 μL of each primer (10 μM), and PCR-grade water to make up the final volume. The Roche Light Cycler 480 96-well plate platform was used for real-time PCR, with cycling conditions starting at 95 °C for 10 min, followed by 35 cycles of 95 °C for 15 s and 60 °C for 1 min. For every PCR, DNA from strain 6BC was used as the positive control, and PCR-grade water was the negative control.

### 2.3. Phylogenetic and Sequence Analysis

To further characterize the genotype of avian chlamydia in birds in Guangxi, the *ompA* gene of positive samples was amplified using nested PCR as previously described [36,37]. Additionally, one to three positive samples of poultry with the highest Ct values per farm from each poultry farming company were selected for further amplification. Given the limited sample size of pet birds and wild birds, all positive samples from these avian groups were included in the amplification process. A 1108-bp segment of the *ompA* gene, encompassing variable domains (VDs) I-IV, was amplified using nested PCR. The outer primers were F2: 5′-GAAAAAACTCTTGAAATCGG-3′ and R2: 5′-ATTSATGTGAGCAGCTCTTT-3′. Initial amplification was carried out in a final reaction volume of 20 μL containing 10 μL of 2× Premix Taq (Ex Taq Version 2.0) (Takara, Beijing, China), 1 μM of each primer, and 2 μL of template DNA. The PCR conditions were as follows: heating at 98 °C for 5 min, followed by 40 cycles of 98 °C for 10 s, 55 °C for 30 s, and 72 °C for 1 min. For the second amplification, the primer pair (F3: 5′-TTACAAGCCTTGCCTGTAGGGA-3′ and R3: 5′-ATTAAGCGTGCTTCACCAGT-3′) was used in the same conditions as the first step. The PCR products were purified as previously described. All PCR products were identified by Sangon Biotech Co., Ltd. (Shanghai, China). DNA sequences were compiled and edited using the SEQMAN program of Lasergene7.1 (DNASTAR, USA). Subsequently, low-quality sequences were filtered out, and the remaining 74 representative sequences were selected for phylogenetic tree construction. The nucleotide sequences of the *ompA* gene from this study have been submitted to the GenBank database under the following accession numbers (PV211935-PV212008) (Table 1). Nucleotide sequence alignment was performed using the MegAlign module of DNASTAR software. The phylogenetic tree was constructed using MEGA 11 software, employing the maximum likelihood method with the General Time Reversible model and bootstrap analysis (1000 replicates). Horizontal distances in the tree are proportional to genetic distance. The phylogenetic tree was annotated and visualized using the online tool EvolView (accessible at https://evolgenius.info//evolview-v2, accessed on 10 September 2025). The nucleotide sequences of the *ompA* gene from reference strains, which shared high similarity to those identified in this study, were selected using the BLAST tool (BLAST 2.17.0) provided by the National Center of Biotechnology Information (NCBI). Additionally, classical strains representative of each genotype were included. The reference sequences of the *ompA* gene are listed in Appendix A.

### 2.4. Ethics Statement

Samples of live animals in this study were collected by certified veterinarians Zhong, H.T. (Veterinary License Number: A012021450095) and Liang, Y.J. (Veterinary License Number: A012023350270) for molecular identification. All work in this study did not involve animal and/or human subjects and was approved by the biosafety committee of Guangxi University (GXUKE2021-02; 16 March 2021). All experimental protocols were approved by the Animal Ethics Committee of Guangxi University (Guangxi, China), with the Animal Experimental Ethics Review Number being “GXU-2023-108”.

### 2.5. Statistical Analysis

The positive rate of avian chlamydia among birds was analyzed using R software version 4.0.5 [38], with results denoted as 95% confidence intervals (CI) calculated using the Wilson score interval. Statistical differences among groups of poultry, pet birds, and wild birds were determined through the chi-square test.

## 3. Results

### 3.1. Prevalence of Avian Chlamydia in Guangxi

To investigate the prevalence of avian chlamydia in poultry, the combined cloacal and oropharyngeal swabs were collected from chicken, duck, goose, and pigeon farms in Guangxi. Among the chicken samples, 25.05% (95% CI, 21.25–29.23%) tested PCR-positive, with positivity rates of 20.95% (95% CI, 16.70–25.94%) in breeder farms, 11% (95% CI, 6.25–18.63%) in broiler farms, and 47.83% (95% CI, 5.16–11.24%) in layer farms. The positivity rate in layer chickens was significantly higher than in breeder chickens (χ^2^ = 29.52, *p* < 0.001) and broiler chickens (χ^2^ = 34.11, *p* < 0.001). Similarly, the positivity rate for layer duck samples was 34% (95% CI, 25.46–43.72%), significantly exceeding the rates in breeder duck farms (13.89%, 95% CI, 9.59–19.70%; χ^2^ = 15.64, *p* < 0.001) and meat duck farms (7.67% (95% CI, 5.16–11.24%; χ^2^ = 42.56, *p* < 0.001). The positivity rate was 62.30% (95% CI, 53.99–66.18%) in meat pigeon farms and 18.12% (95% CI, 12.93–24.82%) in breeder goose farms (Table 2). Chi-square tests were performed to compare the positivity rates of the poultry groups. The results revealed the positivity rate for pigeons was significantly higher than that for chickens (χ^2^ = 98.83, *p* < 0.001), ducks (χ^2^ = 195.91, *p* < 0.001), and geese (χ^2^ = 76.21, *p* < 0.001). Therefore, pigeons exhibited the highest positivity rate for avian chlamydia, followed by chickens, geese, and ducks.

For pet birds, swabs were collected from 15 species belonging to eight orders, with an overall positivity rate of 40.35% (95% CI, 34.20–46.83%). The positivity rate was significantly higher in birds from the pet market (50.71%, 95% CI, 41.76–60.26%), compared to those from the zoo (23.86%, 95% CI, 16.17–33.74%; χ^2^ = 16.19, *p* < 0.001). Among these samples, PCR-positive results were found in 14 species. In the zoo, PCR-positive swabs were detected in 19.61% (95% CI, 11.85–31.56%) of Anseriformes, 7.14% (95% CI, 1.26–31.47%) of Ciconiiformes, 63.64% (95% CI, 35.48–84.83%) of Galliformes, 22.22% (95% CI, 6.29–54.78%) of Gruiformes, and 33.33% (95% CI, 7.29–80.14%) of Pelecaniformes. Notably, the positivity rate in Galliformes was significantly higher than that of Anseriformes (χ^2^ = 8.89, *p* < 0.05), Ciconiiformes (χ^2^ = 9.44, *p* < 0.05) (Table 3). In the pet market, PCR-positive swabs were identified in 32.26% (95% CI, 21.95–44.64%) of Galliformes, 76.19% (95% CI, 54.93–89.34%) of Passeriformes, and 61.40% (95% CI, 48.42–72.96%) of Psittacidae. The positivity rate for Galliformes was significantly lower than that for Passeriformes (χ^2^ = 12.32, *p* < 0.001) and Psittacidae (χ^2^ = 10.17, *p* < 0.05) (Table 3). Overall, avian chlamydia was found to be circulating in most pet birds.

Samples of wild birds from 10 species across six orders were obtained from a pet hospital, revealing an overall positivity rate of 42.86% (95% CI, 24.52–61.83%). PCR-positive results were found in 66.67% of red turtle doves (95% CI, 29.45–98.68%; Columbiformes) and oriental turtle doves (95% CI, 20.68–98.68%), 50% of cinereous vultures (95% CI, 9.46–90.53%; Accipitriformes) and besras (95% CI, 9.46–90.53%; Falconiformes), and 33.33% of spotted doves (95% CI, 6.49–78.68%; Columbiformes). Conversely, negative samples were observed in savannah nightjars (95% CI, 0–79.37%; Caprimulgiformes), peregrine falcons (95% CI, 0–79.37%; Falconiformes), great cormorants (95% CI, 0–79.37%; Pelecaniformes), egrets, and eastern grass owls (95% CI, 0–79.37%; Strigiformes) (Table 4). Chi-square tests were employed to assess the positivity rates among the groups of wild birds, with results indicating no significant differences.

### 3.2. Genetic Characteristics of the Positive Samples

To investigate the genotypes of avian chlamydia in birds, the *ompA* gene from 74 representative positive samples was used to genetic analysis. Phylogenetic analysis showed that most positive samples from chicken farms belonged to the *C. gallinacea* group (Figure 2). However, two positive samples from layer chickens were classified into the *C. psittaci* genotype B group. The nucleotide sequences of pigeon samples were all classified into *C. psittaci* genotype B and were closely related to the representative genotype B strain CP3. By contrast, the nucleotide sequences of goose samples were first clustered into *C. psittaci* genotype A, with one sample being closely related to strains P5 and D121, which were associated with human psittacosis cases reported in China in 2020. The positive samples from ducks were assigned to *C. psittaci* genotypes A and B and the *C. gallinacea* group. Interestingly, two positive isolates from meat ducks, derived from *C. psittaci* genotype A, were also closely related to the strains P5 and D121. These phylogenetic data suggest that the prevalence of avian chlamydia in waterfowl may increase the risk of spillover infections from poultry to humans. Furthermore, six positive duck samples were first classified into *C. psittaci* genotype B group in China, and were closely related to the strain CP3. Therefore, cross-transmission between waterfowl and other birds may have occurred in Guangxi. Overall, these results indicate that various genotypes of avian chlamydia are prevalent in poultry in Guangxi, and cross-transmission may broaden the host range of avian chlamydia and increase cross-species transmission risk.

For pet birds, the Anseriformes-associated and Galliformes-associated samples from the zoo were classified into *C. psittaci* genotype A and the *C. gallinacea* group (Figure 2). The Pelecaniformes-associated samples were only clustered into the *C. gallinacea* group. Moreover, the positive samples from the pet market included three groups: *C. psittaci* genotypes A and B and *C. gallinacea*. Most of the Galliformes positives were clustered into the *C. gallinacea* group and were closely related to strains circulating in China and South Korea. However, one sample from a blue-breasted quail (Galliformes) was classified as *C. psittaci* genotype A, showing a close genetic correlation with the representative strain VS1. In contrast, the majority of Psittacidae-associated positive samples belonged to genotype A, with one sample from a budgerigar clustered into genotype B. Additionally, the nucleotide sequences obtained from five positive samples of Passeriformes were successfully identified and classified into the *C. gallinacea* group. These results suggest that co-circulation of multiple genotypes occurs within pet birds in Guangxi. Notably, *C. gallinacea* was reported in pet birds in China for the first time, highlighting the genetic diversity of avian chlamydia in these populations.

For wild birds, positive samples from the pet hospital were classified into two groups: *C. psittaci* genotypes A and B (Figure 2). The Falconiformes-associated positive samples belonged to genotype A and were closely related to strains circulating in Argentina. Most of the Columbiformes-associated positives were clustered into genotype A, but one positive sample from the red turtle dove (Columbiformes) belonged to genotype B.

## 4. Discussion

Birds, including domestic poultry, companion birds, and wild birds, serve as reservoirs for avian chlamydia [1,5]. To investigate the prevalence of avian chlamydia in birds in Guangxi province, cloacal and/or oropharyngeal swabs were collected from healthy birds in poultry farms, pet markets, a pet hospital, and a zoo. Based on a meta-analysis, a systematic study estimated that the global prevalence of chlamydial infections in birds is 19.5% (95% CI, 16.3–23.1%), with 18.4% (95% CI, 12.5–26.4%) in Asia [39]. Most of the infection reports are associated with *C. psittaci*. In this study, the total positivity rate of avian chlamydia was 28.20% (95% CI, 27.58–28.90%, 492/1744), which is higher than the global and Asian averages.

Among domestic poultry, chickens were considered less sensitive to *C. psittaci* before 2000. However, reports of *C. psittaci* in chickens have emerged in several countries, with positivity rates ranging from 6.9% to 96% [40,41,42,43]. Since 2009, a new avian chlamydia (*C. gallinacea*) has been identified, primarily spreading in chickens in many countries, showing positivity rates from 13.6% to 81.2% [20,21,22,23,24]. *C. abortus*, commonly found in pigeons and psittacines, was first detected in chickens in 2017 with a positivity rate of 15.4% [40,44]. In this study, the overall positivity rete of avian chlamydia in chickens was 25.05% (95% CI, 21.25–29.23%), comparable to the rate observed in commercial chickens in Belgium (23.3%), but lower than the global prevalence of chlamydia in Galliformes (32.0%, 95% CI, 20.6–46.1%) [39,40]. Furthermore, the DNA-positivity rate of avian chlamydia was significantly higher than in layer chickens compared to breeder chickens (χ^2^ = 29.52, *p* < 0.001) and broiler chickens (χ^2^ = 34.11, *p* < 0.001). In Belgium, *C. psittaci* was found in 94%, 100%, and 94% of culture-positive samples from layer, broiler, and breeder farms, respectively, with associated reports of respiratory symptoms and mortality. Yin et al. reported that *C. psittaci*-seropositive results in 90% of layer farms, 96% of broiler farms, and 93% of breeder farms [42]. Nevertheless, direct comparisons of positivity rates between these studies are challenging due to the different detection methodologies. Previous studies have revealed that vertical transmission can occur in chickens via the penetration of *C. gallinacea* from the eggshell into the albumen, yolk, and growing embryo [20,45]. This eggshell penetration leads to transmission to hatching chicks and broilers. Therefore, the effective cleaning of eggs contaminated with avian chlamydia should be considered a critical control step to prevent pathogen transmission in chickens. To understand the genotypes of avian chlamydia in chickens, the *ompA* gene sequences from 19 representative positive samples were sequenced and then used for phylogenetic analysis, consistent with previous studies [12,14]. Genotyping based on the *ompA* gene in previous studies has revealed the presence of *C. psittaci* genotypes A, B, C, D, E, F, and E/B, as well as *C. gallinacea,* in chickens [33,40,41,42,43,46]. These reports predominantly detected *C. psittaci* genotypes B and D in chickens. In contrast, in the present study, *C. gallinacea* was the most frequently identified species in positive samples from chicken farms, consistent with a nationwide detection that *C. gallinacea* has become an endemic chlamydial species in chickens in China [20]. Additionally, a wide range of genotypes, including *C. psittaci* genotypes A, B, C, D, F, and *C. gallinacea,* have been documented in chickens across China [20,42,46,47]. Hence, the avian chlamydia identified in Chinese chicken flocks is characterized by the co-circulation of mutiple diverse genotypes.

In domestic waterfowl, ducks are considered particularly susceptible to *C. psittaci* infection, which can lead to significant economic losses in duck farms due to reduced egg production [48,49]. In this study, the positivity rate of avian chlamydia in ducks was 14.14% (95% CI, 11.57–17.26%, 82/580), which is lower than the rates reported in Egypt (80%, 78/97), France (40%, 95% CI, 12–77%) and Taiwan (21.6%, 134/620) [48,50,51]. Variations in positivity rates between these studies may be associated with the differences in duck species and sampling locations. Notably, samples from Egyptian duck flocks were collected from birds suffering from diarrhea, debilitation, anorexia, respiratory symptoms, and emaciation [51]. In contrast, *C. psittaci*-seropositive results were observed in 38.92% (130/334) of market-sold adult ducks and 20.9% (56/268) of domestic ducks in China [52,53]. The lower DNA-positivity rate of ducks seen in this study may be due to the intermittent shedding of the pathogen, which leads to an underestimation of infection when detected by PCR. Furthermore, the positivity rate of samples from layer duck farms was significantly higher compared to those from breeder duck farms (χ^2^ = 15.64, *p* < 0.001) and broiler duck farms (χ^2^ = 42.56, *p* < 0.001) in this study. The varying susceptibility to avian chlamydia infection among these duck flocks needs further investigation. Regarding host or species-specific genotypes, *C. psittaci* genotypes C, E/B, and E have primarily been isolated from ducks [4,31]. Nevertheless, a broader range of avian chlamydia genetypes has been identified in ducks populations. Notably, a mixed infection involving *C. psittaci* genotypes A and C was detected in wild mallards in New Zealand [54]. In China, *C. psittaci* genotype A has been associated with a decline in egg production decline in laying ducks [49]. Additionally, *C. gallinacea* has been detected in duck flocks in France, marking its first report in Chinese duck flocks [19]. Recently, isolates from ducks in Poland have been clustered with genotypes G1 and G2, which are considered intermediates between *C. psittaci* and *Chlamydia abortus* [44]. Here, *C. psittaci* genotypes A and B, as well as to the *C. gallinacea* were identified in ducks flocks, suggesting that ducks may serve as hosts for multiple avian chlamydia genotypes. Notably, genotype B, which is predominantly detected in pigeons, was initially reported in ducks in China. There are limited reports documenting the presence of *C. psittaci* genotype B in ducks. However, in France, an isolate from ducks linked to human cases of psittacosis seems to be an intermediate between genotypes A and B [55].

In studies investigating chlamydia infection in ducks, the seroprevalence of chlamydia infection in geese was reported to be 22% in southern China and 19.0% in northeastern China [53,56]. Furthermore, *C. psittaci* was isolated from pharyngeal swabs of 58% (47/81) of Canada geese without clinical symptoms in Belgium [57]. In this study, the positivity rate of avian chlamydia in goose was 18.12% (95% CI, 12.93–24.82%), which is lower than previously reported rates but higher than the positivity rates found in goose flocks in Poland (5.6%, 1/18) and Taiwan (6.0%, 5/83) [48,58]. In geese, *C. psittaci* genotypes B, C, E, and E/B have been identified [4,57,59]. Notably, *C. psittaci* genotype A was first detected in geese in this study. Waterfowl frequently share open water habitats with wild birds, thereby increasing the potential for cross-transmission between these populations. Additionally, recent studies have indicated that human infections with *C. psittaci* in China are associated with genotype A strains circulating among duck flocks and genotype E/B strains found in geese [59,60]. Phylogenetic analysis revealed that two samples from ducks and one from a goose were closely related to strains (P5 from patients and D121 from ducks, respectively) associated with human psittacosis cases in China, with *ompA* sequence similarities between those strains ranging from 99.8% to 100% (Appendix A) [60]. Consequently, domestic waterfowl act as significant reservoirs for the transmission of avian chlamydia to humans. However, the lack of swab samples from workers at the duck and goose farms due to personal preferences and ethical considerations hindered the assessment of the zoonotic potential of avian chlamydia. Hence, it is essential to maintain continued surveillance and implement effective management strategies for avian chlamydia in waterfowl to reduce zoonotic transmission.

As is widely known, pigeons are considered to have the highest rates of chlamydia infections among birds, and raising pigeons has been proposed as a risk for zoonotic transmission. Human infection with *C. psittaci* is commonly acquired through close contact with feral pigeons, often during feeding activities. Consequently, numerous studies on avian chlamydia infection have focused on feral pigeons [4,5]. Although data on avian chlamydia in domestic pigeons are limited, recent studies conducted in China have reported positivity rates ranging from 0.75% (1/132) to 46.67% (7/15) [47,61,62]. Conversely, the positivity rate of avian chlamydia in meat pigeons in Guangxi was 62.30% (95% CI, 53.99–66.18%), which is lower than that of Egypt (86.7%), but higher than those reported for domestic pigeons in Germany (12.8–42.6%, 14/109–58/136), Iraq (40%, 60/150) and Switzerland (14.7%%, 5/34) [51,63,64,65]. The factors contributing to avian chlamydia infections in homing pigeons were likely associated with their interactions with rodents and wild animals, and management practices. Additionally, it has been demonstrated that pigeons co-infected with both *C. psittaci* and PiCV (pigeon circovirus) had infection rates 2–3 times higher than those infected with *C. psittaci* alone. This suggests that the shedding of chlamydia by pigeons is intermittent and may be activated by immunosuppression due to co-infections such as PiCV [66,67]. Consequently, variables such as the environmental conditions of pigeon aviaries and co-infection with other diseases may influence the prevalence of *C. psittaci* in pigeons, warranting further study to enhance understanding of these dynamics. Furthermore, our study indicated that the *ompA* gene from pigeon samples was classified into *C. psittaci* genotype B group. This finding aligns with previous research, where genotype B was the most frequently isolated genotype from pigeons, although other genotypes, including A, C, D, and E, have been detected [1,4]. Notably, *C. avium*, a newer avian chlamydia species reported in pigeons from France, Germany, Italy, and the Netherlands [15,68,69], was not detected in this study.

Human psittacosis cases are often linked to exposure to infected pet birds in homes, pet shops, bird fairs, markets, zoos, and parks [5,10]. The avian chlamydia positivity rate in pet birds was 40.35% (95% CI, 34.20–46.83%, 92/228), surpassing rates reported in Belgium (39.3%, 33/84), Argentina (30%, 27/90), Japan (7.2%, 48/668), and Hong Kong (0.97%, 5/516), but lower than in Korea (63.9%, 168/263) [70,71,72,73]. These variations in positivity rate can be attributed to differences in bird species, age, gender, season, location, and diagnostic methods used. In this study, pet market birds (50.71%, 95% CI, 41.76–60.26%) showed a significantly higher positivity rate compared to zoo birds (23.86%, 95% CI, 16.17–33.74%; χ^2^ = 16.19, *p* < 0.001), consistent with prior findings [9]. Conditions prevalent in pet markets, such as group and mixed feeding, stacked cages, inadequate ventilation, and delayed cleaning of fecal debris or spilled food, likely contribute to elevated infection rates. Within the zoo, the positivity rate in Galliformes (63.64%, 95% CI, 35.48–84.83%) was significantly higher compared to Anseriformes (19.61%, 95% CI, 11.85–31.56%) and Ciconiiformes (7.14% 95% CI, 1.26–31.47%). A comprehensive global analysis of avian chlamydia prevalence corroborates these findings, indicating higher infection rates in Galliformes (32%, 95% CI, 20.6–46.1%) than in Anseriformes (20.6%, 95% CI, 12.7–31.4%) [39]. In the pet market, the positivity rate in Galliformes was significantly lower than that in Passeriformes and Psittacidae. Previous studies predominantly focused on chlamydia infections in Psittacidae, with positivity rates ranging from 19.9% to 63.9% [9,70,72]. The discrepancy may be attributed to the limitation of sample size and geographical scope. Phylogenetic analysis identified *C. psittaci* genotypes A and B, as well as *C. gallinacea,* in pet birds. In China, *C. psittaci* genotypes A, B, and CPX0308 have been reported in pet birds [9,47,73,74]. Notably, *C. gallinacea* was detected in pet birds in China for the first time. These findings suggest a diverse range of avian chlamydia genotypes present in pet birds in Guangxi, potentially linked to factors such as mixed feeding practices, close contact during immune challenges, and the introduction of birds from other zoos or intermediaries that may have had contact with wild birds. Consistent with our findings, *C. psittaci* genotypes A and B have been primarily identified in the family Psittacidae. However, recent reports have also documented the presence of *C. gallinacea, C. abortus,* and *C. avium* within this group [21,70,75]. The widespread dissemination of avian chlamydia among pet bird populations in Guangxi highlights a zoonotic hazard. Therefore, attention should be allocated to the regulation and management of the pet bird trade.

Wild birds are recognized as natural reservoirs of avian chlamydia. Transmission can occur to poultry, pet birds, and humans through direct contact with infected birds or their excretions [5,10]. Recently, avian chlamydia infections have been identified in wild birds across Europe, Asia, North America, South America, and Africa, with positivity rates varying from 0.9% to 29% [5]. In the present study, the positivity rates in wild birds were 42.86% (95% CI, 24.52–61.83%), which was higher than the rates reported in previous studies. *C. psittaci* infection is globally distributed in wild birds, with Columbiformes, especially feral pigeons, being common hosts. Recent studies have also identified Corvidae and Accipitridae as significant hosts [5]. A global meta-analysis revealed that Galliformes had the highest prevalence (32%), while Passeriformes showed the lowest (13.4%), which contrasts with this study [39]. Nevertheless, the ability to make comparisons is constrained by the lack of comprehensive data on the true prevalence of chlamydial infections in wild bird populations, primarily due to insufficient surveillance efforts. Additionally, the collection of wild bird samples predominantly from veterinary submissions, community reports, or wildlife rehabilitation centers introduces a potential sampling bias. Phylogenetic analysis indicated *C. psittaci* genotypes A and B in Columbiformes, whereas genotype A was identified in Falconiformes. *C. psittaci* has been detected in a wide range of wild birds, with several genotypes reported. Genotype A is usually identified in multiple avian orders, including Psittaciformes, Passeriformes, Accipitriformes, Falconiformes, and Procellariiformes [76,77]. In contrast, genotypes B and E are common in Columbiformes, particularly pigeons [78]. The genotypes identified in our study are in accordance with these reports. Furthermore, new avian chlamydia species have been described in wild birds. For instance, *C. gallinacea* has been found in Psittaciformes and Charadriiformes in Australia and Korea [26,77]. *C. avium* has been detected in Columbiformes in Switzerland and the Netherlands, and a ring-necked parakeet (Psittaciformes) in France [64,68]. Moreover, *C. ibidis* has been isolated from a crested ibis (Pelecaniformes) in China [18]. However, these emerging species were not found in our study.

Our study faced several limitations. Firstly, the ambiguity surrounding the age and sampling season of the birds hindered our ability to evaluate potential risk factors associated with the prevalence of avian chlamydia. Secondly, the study’s sample composition was skewed, as poultry samples were predominantly obtained from captive birds, whereas samples from pet and wild birds were limited. This imbalance obscured the potential transmission pathways of avian chlamydia from pet or wild birds to poultry. Thirdly, the real time PCR methodology employed was insufficient for distinguishing among all avian chlamydia species, thereby constraining our analysis of the true prevalence of various species. Fourthly, the assessment of the zoonotic potential of avian chlamydia in Guangxi was impeded by the absence of human samples. Lastly, the amplification of the *ompA* gene fragment was restricted to 1108-bp, indicating that a more comprehensive analysis of genetic diversity should be pursued using alternative methodologies. Moreover, there is an urgent imperative to develop advanced molecular detection methods capable of both identifying and differentiating among the various avian chlamydia species.

## 5. Conclusions

This study demonstrated a widespread distribution of avian chlamydia among bird populations in Guangxi. The genetic analysis of positive samples indicated that avian chlamydia exhibited genetic diversity among bird populations, with an expanding host range suggesting its potential for cross-species transmission. Consequently, continuous surveillance of avian chlamydia in bird populations and monitoring its genetic evolution characteristics are crucial to reduce public health risks.

## Figures and Tables

**Figure 1 microorganisms-13-02220-f001:**
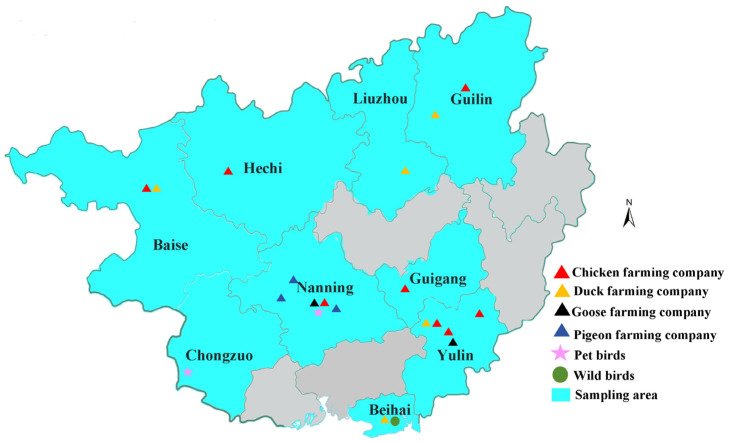
The regions of bird sampling in Guangxi, southwestern China. The Guangxi standard map was obtained from the Standard Map Service System (online website: https://bzdt-ch-mnr-gov-cn.vpn.gxu.edu.cn:8118/, accessed on 10 September 2025). Sampling locations were processed using Adobe Photoshop CC 2018 software. Poultry farming companies are represented by triangles, with different colors indicating various avian species: red for chickens, orange for ducks, black for geese, and blue for pigeons. Pet bird samples are represented by stars in magenta, while additional wild bird samples are denoted by circles in green.

**Figure 2 microorganisms-13-02220-f002:**
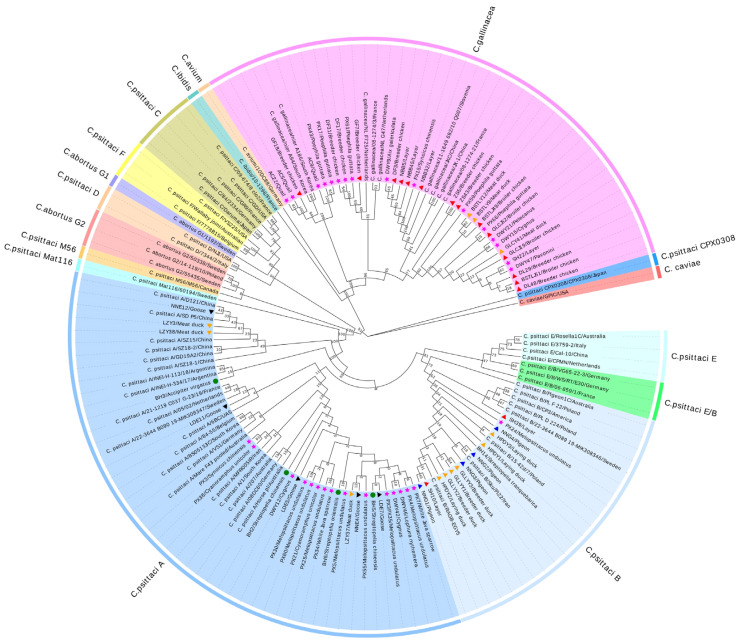
Phylogenetic tree based on nucleotide sequences of the *ompA* gene. The *ompA* gene of 74 representative positive samples obtained in this study and various genotypes of *C. psittaci* and other *Chlamydia* spp. available in GenBank. The accession numbers and host information for the samples are detailed in Appendix A. In the phylogenetic tree, poultry samples are indicated by triangles, with specific colors denoting different avian species: red for chickens, orange for ducks, black for geese, and blue for pigeons, respectively. Pet bird samples are represented by stars, marked in magenta, while additional wild bird samples are denoted by circles, marked in green.

**Table 1 microorganisms-13-02220-t001:** The nucleotide sequences of the *ompA* gene used in this study.

Positive Sample	Source	Host	Location	GenBank No.
DF3	Chicken farm-DF-A	Breeder chicken	Guilin	PV211947
DF17	Chicken farm-DF-A	Breeder chicken	Guilin	PV211948
DF31	Chicken farm-DF-B	Breeder chicken	Guilin	PV211949
DL29	Chicken farm-DL-A	Breeder chicken	Hechi	PV211950
DL48	Chicken farm-DL-B	Breeder chicken	Hechi	PV211951
GF7	Chicken farm-GF-A	Breeder chicken	Guigang	PV211959
GF18	Chicken farm-GF-B	Breeder chicken	Guigang	PV211960
ZS5	Chicken farm-ZS-A	Breeder chicken	Yulin	PV212007
ZS43	Chicken farm-ZS-B	Breeder chicken	Yulin	PV212008
GLCJ19	Chicken farm-GLC-A	Bioiler chcken	Yulin	PV211961
GLCJ52	Chicken farm-GLC-B	Bioiler chicken	Yulin	PV211962
BSTLJ49	Chicken farm-TL-A	Bioiler chicken	Baise	PV211943
BSTLJ51	Chicken farm-TL-B	Broiler chiken	Baise	PV211944
NBB5	Chicken farm-NBB-A	Layer chicken	Nanning	PV211976
NBB32	Chicken farm-NBB-B	Layer chicken	Nanning	PV211977
NBB45	Chicken farm-NBB-B	Layer chicken	Nanning	PV211978
SH10	Chicken farm-SH-A	Layer chicken	Yulin	PV212004
SH22	Chicken farm-SH-A	Layer chicken	Yulin	PV212005
SH39	Chicken farm-SH-B	Layer chicken	Yulin	PV212006
GLLYY1	Duck farm-LY-A	Breeder mallard duck	Guilin	PV211963
GLLYY2	Duck farm-LY-B	Breeder mallard duck	Guilin	PV211965
GLLYY3	Duck farm-LY-C	Breeder cherry valley duck	Guilin	PV211966
BSTLY12	Duck farm-TL-B	Meat duck	Baise	PV211945
BSTLY9	Duck farm-TL-A	Meat duck	Baise	PV211946
GLCY41	Duck farm-GLC-B	Meat duck	Yulin	PV211964
LZY3	Duck farm-LZ-A	Meat duck	Liuzhou	PV211973
LZY37	Duck farm-LZ-B	Meat duck	Liuzhou	PV211974
LZY38	Duck farm-LZ-C	Meat duck	Liuzhou	PV211975
HPDY1	Duck farm-HP-A	Layer duck	Beihai	PV211967
HPDY2	Duck farm-HP-B	Layer duck	Beihai	PV211968
HPDY3	Duck farm-HP-D	Layer duck	Beihai	PV211969
LDE5	Goose farm-LYLE-A	Meat goose	Nanning	PV211970
LDE7	Goose farm-LYLE-B	Meat goose	Nanning	PV211971
LDE11	Goose farm-LYLE-C	Meat goose	Nanning	PV211972
NNE4	Goose farm-NNST-B	Meat goose	Nanning	PV211979
NNE12	Goose farm-NNST-C	Meat goose	Nanning	PV211980
NNG1	Pigeon farm-HP-A	Meat pigeon	Nanning	PV211981
NNG2	Pigeon farm-HP-B	Meat pigeon	Nanning	PV211982
NNG3	Pigeon farm-LT-E	Meat pigeon	Nanning	PV211983
NNG4	Pigeon farm-XR-B	Meat pigeon	Nanning	PV211984
AC5	Pet market	Quail (Galliformes)	Chongzuo	PV211935
AC9	Pet market	Quail (Galliformes)	Chongzuo	PV211936
AC22	Pet market	Quail (Galliformes)	Chongzuo	PV211937
PX3	Pet market	Blue-breasted Quail (Galliformes)	Chongzuo	PV211985
PX4	Pet market	Budgerigar (Psittacidae)	Chongzuo	PV211986
PX5	Pet market	Budgerigar (Psittacidae)	Chongzuo	PV211987
PX7	Pet market	White Java Sparrow (Passeriformes)	Chongzuo	PV211988
PX15	Pet market	Blue-breasted Quail (Galliformes)	Chongzuo	PV211989
PX17	Pet market	Zebra Finch (Passeriformes)	Chongzuo	PV211990
PX21	Pet market	Antipodes Green Parakeet (Psittacidae)	Chongzuo	PV211991
PX24	Pet market	Budgerigar (Psittacidae)	Chongzuo	PV211992
PX25	Pet market	Budgerigar (Psittacidae)	Chongzuo	PV211993
PX30	Pet market	Budgerigar (Psittacidae)	Chongzuo	PV211994
PX34	Pet market	White Java Sparrow (Passeriformes)	Chongzuo	PV211995
PX35	Pet market	Budgerigar (Psittacidae)	Chongzuo	PV211996
PX38	Pet market	Antipodes Green Parakeet (Psittacidae)	Chongzuo	PV211997
PX55	Pet market	Budgerigar (Psittacidae)	Chongzuo	PV211998
PX43	Pet market	Zebra Finch (Passeriformes)	Chongzuo	PV211998
PX59	Pet market	Zebra Finch (Passeriformes)	Chongzuo	PV212000
PX60	Pet market	Budgerigar (Psittacidae)	Chongzuo	PV212001
PX61	Pet market	Zebra Finch (Passeriformes)	Chongzuo	PV212002
PX66	Pet market	Zebra Finch (Passeriformes)	Chongzuo	PV212003
DWY9	Zoo	Mandarin Duck (Anseriformes)	Nanning	PV211952
DWY12	Zoo	Swan (Anseriformes)	Nanning	PV211953
DWY15	Zoo	Swan (Anseriformes)	Nanning	PV211954
DWY21	Zoo	Pelican (Pelecaniformes)	Nanning	PV211955
DWY42	Zoo	Swan (Anseriformes)	Nanning	PV211956
DWY46	Zoo	Silver Pheasant (Galliformes)	Nanning	PV211957
DWY47	Zoo	Peafowl (Galliformes)	Nanning	PV211938
BH2	Pet hospital	Oriental Turtle-dove (Columbiformes)	Beihai	PV211939
BH3	Pet hospital	Besra (Falconiformes)	Beihai	PV211940
BH5	Pet hospital	Oriental Turtle-Dove (Columbiformes)	Beihai	PV211941
BH6	Pet hospital	Oriental Turtle-Dove (Columbiformes)	Beihai	PV211942
BH14	Pet hospital	Red Turtle Dove (Columbiformes)	Beihai	PV211938

**Table 2 microorganisms-13-02220-t002:** The positivity rate of avian chlamydia in poultry.

Source	Farm	Positivity Rate (95% CI, Positive Samples/Total Samples)	
Chicken ^b^	Breeder chicken	20.95% (95% CI, 16.70–25.94%, 62/296)	25.05% (95% CI, 21.25–29.23%, 128/511)
	Broiler chicken	11% (95% CI, 6.25–18.63%, 11/100)	
	Layer chicken	47.83% (95% CI, 38.92–56.88%, 55/115)	
Duck ^c^	Breeder duck	7.67% (95% CI, 5.16–11.24%, 23/300)	14.14% (95% CI, 11.57–17.26%, 82/580)
	Meat duck	13.89% (95% CI, 9.59–19.70%, 25/180)	
	Layer duck	34% (95% CI, 25.46–43.72%, 34/100)	
Goose ^bc^	Breeder goose	18.12% (95% CI, 12.93–24.82%, 29/160)	18.12% (95% CI, 12.93–24.82%, 29/160)
Pigeon ^a^	Meat pigeon	62.30% (95% CI, 55.37–68.69%, 152/244)	62.30% (95% CI, 55.37–68.69%, 152/244)
Total	26.16% (95% CI, 23.98–28.45%, 391/1495)		

The positivity rates were denoted with Wilson 95% CI. Statistically significant differences among poultry groups were evaluated using the chi-square test, with different letters (a, b, c) indicating groups of significant differences.

**Table 3 microorganisms-13-02220-t003:** The positivity rate of avian chlamydia in pet birds.

Source	Order	Species	Positivity Rate (95% CI, Positive Samples/Total Samples)	
Zoo	Anseriformes ^b^	Swan	18.75% (95% CI: 10.00–31.23%, 9/48)	19.61% (95% CI, 11.85–31.56%) 10/51)
		Mandarin Duck	33.33 (95% CI, 3.10–79.30%, 1/3)	
	Ciconiiformes ^b^	White Stork	0% (95% CI, 0–35%, 0/7)	7.14% (95% CI, 1.26–31.47%, 1/14)
		Flamingo	14.29% (95% CI, 3–51%, 1/7)	
	Galliformes ^a^	Peafowl	55.56% (95% CI, 26.57–81.23%, 5/9)	63.64% (95% CI, 35.48–84.83%, 7/11)
		Silver pheasant	100% (95% CI, 34.23–100%, 2/2)	
	Gruiformes ^ab^	White Crane	50% (95% CI, 9.46–90.54%, 1/2)	22.22% (95% CI, 6.29–54.78%, 2/9)
		Red-crowned Crane	14.29% (95% CI, 2.575–51.33%, 1/7)	
	Pelecaniformes ^ab^	Pelican	33.33% (95% CI, 7.29–80.14%, 1/3)	33.33% (95% CI 7.29–80.14%, 1/3)
	Subtotal	23.86% (95% CI, 16.17–33.74%, 21/88)		
Pet market	Galliformes ^b^	Blue-breasted Quail	71.43% (95% CI, 35.90–91.81%, 5/7)	32.26% (95% CI, 21.95–44.64%, 20/62)
		Quail	27.27% (95% CI, 17.28–40.23%, 15/55)	
	Passeriformes ^a^	Zebra finch	77.77% (95% CI, 45.24–93.69%, 7/9)	76.19% (95% CI, 54.93–89.34%, 16/21)
		White Java sparrow	75% (95% CI, 46.76–91.13%, 9/12)	
	Psittacidae ^a^	Budgerigar	58.90% (95% CI, 43.43–72.905, 23/39)	61.40% (95% CI, 48.42–72.96%, 35/57)
		Antipodes Green Parakeet	66.67% (95% CI, 43.76–83.73%, 12/18)	
	Subtotal	50.71% (95% CI, 41.76–60.26%, 71/140)		
Total	40.35% (95% CI, 34.20–46.83%, 92/228)			

The positivity rates were denoted with Wilson 95% CI. Statistically significant differences among poultry groups were evaluated using the chi-square test, with different letters (a, b) indicating groups with significant differences.

**Table 4 microorganisms-13-02220-t004:** The prevalence of avian chlamydia in wild birds.

Source	Order	Species	Positivity Rate (95% CI, Positive Samples/Total Samples)	
Pet hospital	Accipitriformes	Cinereous Vulture	50% (95% CI, 9.46–90.53%, 1/2)	50% (95% CI, 9.45–90.55% 1/2)
	Caprimulgiformes	Savannah Nightjar	0 (95% CI, 0–79.37%, 0/1)	0 (0/1)
	Columbiformes	Red Turtle Dove	66.67% (95% CI, 29.45–98.68%, 4/6)	58.33% (95% CI, 31.90–80.12%, 7/12)
		Oriental Turtle-dove	66.67% (95% CI, 20.68–98.68%, 2/3)	
		Spotted Dove	33.33% (95% CI, 6.49–78.68%, 1/3)	
	Falconiformes	Besra	50% (95% CI, 9.46–90.53%, 1/2)	33.33% (95% CI, 6.14–79.24%, 1/3)
		Peregrine Falcon	0 (95% CI, 0–79.37%, 0/1)	
	Pelecaniformes	Egret	0 (95% CI, 0–79.37%, 0/1)	0 (95% CI, 0–65.76%, 0/2)
		Great Cormorant	0 (95% CI, 0–79.37%, 0/1)	0 (95% CI, 0–79.37%, 0/1)
	Strigiformes	Eastern Grass-owl	0 (95% CI, 0–79.37%, 0/1)	0 (95% CI, 0–79.37%, 0/1)
Total	42.86 (95% CI, 24.52–61.83%, 9/21)			

The positivity rates were denoted with Wilson 95% CI. Statistical evaluation of the significant differences in positivity rates among poultry groups was conducted using the chi-square test. No significant differences were detected among the wild bird groups.

## Data Availability

The nucleotide sequences of the ompA gene in our study have been submitted to the GenBank database under the following accession numbers (PV211935-PV212008) (Appendix A). [GenBank] [https://www.ncbi.nlm.nih.gov/, accessed on 10 September 2025] [PV211935-PV212008].

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
