# Peer review of "Prevalence and Genetic Characteristics of Avian Chlamydia in Birds in Guangxi, Southwestern China"

_microorganisms, 2025, doi:10.3390/microorganisms13092220_

Round 1
Reviewer 1 Report
Comments and Suggestions for Authors
Please see atachmnent file

Author Response
Response to Reviewer 1 Comments
- Summary
We thank the reviewer for the kind consideration and constructive comments on our manuscript. We have carefully revised the manuscript and provided a detailed point-by-point response, with changes highlighted in red. We hope these changes will enhance the quality of our manuscript.
- Questions for General Evaluation
Does the introduction provide sufficient background and include all relevant references?/ Yes
Response: Thank you for your review and your professional suggestions.
Is the research design appropriate? /Must be improved
Response: Thank you for your review and your professional suggestions. We have added the description, table, and figure according to your suggestions into the Materials and Methods section, spanning in lines 100-158 and 207-214, with all modifications clearly marked in red.
Are the methods adequately described?/Must be improved
Response: Thank you for your review and your professional suggestions. We have added the description, table, and figure according to your suggestions into the Materials and Methods section, spanning in lines 100-158 and 207-214, with all modifications clearly marked in red.
Are the results clearly presented? /Must be improved
Response: Thank you for your review and your professional suggestions. We have added the description and calculation (including 95% confidence intervals and Statistical differences) according to your suggestions in Result section in line 223-234, line 242-254, and line 261-270, with marking them in red.
Are the conclusions supported by the results? /Must be improved
Response: Thank you for your review and your professional suggestions. According to your suggestions, we have added the description and calculations including 95% confidence intervals and statistical differences into the Result section on lines 219-230, 237-250, and 257-264, with these additions highlighted in red.
Are all the figures and tables clear and well-presented? /Can be improved
Response: We are grateful for your review and your professional suggestions. We have added the tables and figure according to your suggestions in the Materials and Methods, Results, and Supplementary Materials sections. Table1 found on line 205, while Table S1 and Table S2 have also been added. Figure 1, which denotes the sampling area presented on line 121.
- Point-by-point response to Comments and Suggestions for Authors
Comments 1: I am curious why (and how many) samples were collected from two quite different anatomical sites on the birds (cloacal and oropharyngeal). How many samples came from the cloaca and how many from the oropharynx? Was there any difference in the number of positive samples between these two sampling sites?
Response 1: We sincerely appreciate your riview and constructive suggestions. We are terribly sorry for any confusion caused by the inappropriate description in the Materials and Methods section. According to previous studies[1-5], we designed our this experiment. Previous studies have revealed that avian chlamydia can be found in respiratory tract exudate and fecal material of infected birds. Samples should be collected from both the respiratory and gastrointestinal tracts whenever possible for better accuracy [3-7]. Choanal/oropharyngeal swabs are more consistent for isolation of the agent than fecal swabs, especially during early stages of infection. Given the intermittent shedding of chlamydia, we combined the cloacal and oropharyngeal swabs to enhance diagnostic sensitivity. However, due to concerns from pet owners regarding potential sales-related issues, a total of 140 cloacal swabs was collected exclusively from pet birds in pet market. We have provided the detailed information of samples in Table S1 and added description in Materials and Methods section on lines 101-103 and 156-157, with these additions highlighted in red.
Reference
- Andersen, A.A. Comparison of pharyngeal, fecal, and cloacal samples for the isolation of Chlamydia psittaci from experimentally infected cockatiels and turkeys. J. Vet. Diagn. Invest.1996, 8, 448.
- Fudge, A.M. A review of methods to detect Chlamydia psittaci in avian patients. J. Avian Med. Surg. 1997, 11, 153-165.
- Hulin, V.; Oger, S.; Vorimore, F.; Aaziz, R.; de Barbeyrac, B.; Berruchon, J.; Sachse, K.; Laroucau, K. Host preference and zoonotic potential of Chlamydia psittaci and C. Gallinacea in poultry. Pathog. Dis. 2015, 73, 1-11.
- You, J.; Wu, Y.; Zhang, X.; Wang, X.; Gong, J.; Zhao, Z.; Zhang, J.; Zhang, J.; Sun, Z.; Li, J.; et al. Efficient fecal-oral and possible vertical, but not respiratory, transmission of emerging Chlamydia gallinacea in broilers. Vet. Microbiol. 2019, 230, 90-94.
- 5. Solorzano-Morales, A.; Dolz, G. Molecular characterization of Chlamydia species in commercial and backyard poultry farms in Costa Rica. Infect.2022, 150.
- Vanrompay, D. Avian chlamydiosis. In Diseases of Poultry, 14th ed.; Swayne, D.E, A., Boulianne, M, B., Eds.; John Wiley& Sons, Inc: United States,2020;pp. 1086-1106.
- Soon, X.; Gedye, K.; Benschop, J.; Gartrell, B. Molecular detection of Chlamydia psittaci in birds: a systematic review. Avian Pathology. 2025, 54, 279-298.
Response 2: Thank you for your review and your professional suggestions. We collected 1,604 combined cloacal and oropharyngeal swab samples, along with 140 cloacal swab samples from the pet market. The pooling of cloacal and oropharyngeal swabs was employed as a composite sampling strategy to account for the intermittent shedding of chlamydia. We have provided the detailed information of samples in Table S1.
Response 3: Thank you for pointing this out. We collected a total of 1,604 combined cloacal oropharyngeal swab samples, in addition to 140 cloacal swab samples, from a pet market. Considering the intermittent shedding of chlamydia, we pooled the cloacal and oropharyngeal swabs as a composite sample for detection purposes. Specifically, 140 cloacal swabs were collected exclusively from pet birds in the pet market, as pet owners expressed concerns about potential sales-related issues. The positivity rate for the combined samples was 26.24% (421/1604, 95% CI: 24.14%-28.44%), while the positivity rate for the cloacal swabs samples from the pet market was 50.71% (71/140, 95% CI: 41.76%-60.26%). Additionally, We have provided the detailed information regarding the samples in Table S1. we sincerely hope that you could update tables in the Supplementary Materials if necessary.
Comments 2: What is the product length of the primers used in the nested PCR reaction? Were these primers designed by the authors?
Response 1: Thank you for your review and your professional suggestions. A 1108-bp segment of the ompA gene, encompassing variable domains (VDs) I-IV, was amplified using nested PCR. And we added this description on line in line 180 and highlighted in red.
Response 2: Thank you for pointing this out. Yes, these primers used in this study were designed by Zhong, H.T. and Huang, J.N. And these primers were listed in lines 182 and 187. According to previous studies [9,10], we developed specific primers targeting the conserved regions of the major outer membrane protein A (ompA) genes of Chlamydia psittaci, Chlamydia gallinacea, Chlamydia avium, and Chlamydia ibidis. Furthermore, the specificity, sensitivity and repeatability of these primers have been validated in comparison to the conventional PCR methods[11].
- Geens, T.; Dewitte, A.; Boon, N.; Vanrompay, D. Development of a Chlamydophila psittaci species-specific and genotype-specific real-time PCR. Res. 2005, 36, 787-797.
- Heddema, E.R.; van Hannen, E.J.; Duim, B.; Vandenbroucke-Grauls, C.M.; Pannekoek, Y. Genotyping of Chlamydophila psittaci in human samples. 2006, 12, 1989-1990.
- Zhong, H. Establishment and comparison of three PCR methods for avian chlamydia and its molecular prevalence investigation in some regions of Guangxi during 2021-2023. master's thesis, Guangxi University, Nanning, 2024.
Comments 3: Were all samples that tested positive by RT-PCR also confirmed by nested PCR? And actually, how many samples was positive, in total?
Response 1: Thank you for your review and your professional suggestions. In this study, one to three positive samples per farm, which exhibited highest Ct values and tested positive via RT-PCR, were selected from each poultry farming company for further amplification using nested PCR. Given the limited sample size of pet birds and wild birds, all positive samples from these groups were included in the amplification process. We have added this description on lines 174-180, and highlighted in in red.
Response 2: Thank you for pointing this out. For the chickens samples, sixty-two positive samples that tested positive via RT-PCR were selected for further analysis by nested PCR, resulting in forty-eight positive outcomes. Among the ducks samples, nested PCR testing yielded positive results for forty-one out of fifty-nine selected samples. In the case of geese, fourteen out of twenty-one selected samples tested positive by nested PCR. Additionally, for pigeons, nested PCR testing identified positive results in twenty-one out of thirty selected samples. In the group of pet birds, forty-eight out of ninety-two selected tested positive by nested PCR. Finally, for wild birds, six out of nine selected samples tested positive by nested PCR. The discrepancies between the two methods can be attributed to: (1) variations in sensitivity; (2) differences in primer design; and (3) amplification efficiency.
Comments 4: What model was used to construct the phylogenetic tree, and why did the authors use MEGA 5 software, when MEGA 11—with much more advanced algorithms—available for free?
Response 1: Thank you for your review and your professional suggestions. The phylogenetic tree was reconstructed using MEGA 11 software, employing the maximum likelihood method with the General Time Reversible model and bootstrap analysis (1,000 replicates). Subsequently, the phylogenetic tree was annotated and visualized using the online tool EvolView (available at https://evolgenius.info//evolview-v2). We have added this description on lines 195-202, highlighting them in red.
Comments 5:Table 1 is unnecessary in the main text and should be moved to the supplementary materials.
Response 1: Thank you for your review and your professional suggestions. We have moved this table in supplementary materials Table S2.
Comments 6: Please remove “.1” at the end of all sequence submission numbers throughout the manuscript.
Response 1: Thank you for your review and your professional suggestions. We have removed “.1” at the end of all sequence submission numbers throughout the manuscript.
Comments 7: One of the major weaknesses of the manuscript is the Ethics Statement, which I find particularly unclear. All samples were reportedly collected by a single (!) veterinarian. Please provide this person’s full name and professional identification number (if applicable). Or is one of the authors a veterinarian? Please also provide the name and location of the animal hospital, as well as details for the zoo, pet bird market, and duck, pigeon, and chicken farms. Additionally, please include the biosafety committee permission number from Guangxi University. Even with this approval, I am not convinced that other approval (e.g., from a national veterinary ethics committee) would not be required.
Response 1: Thank you for your review and your professional suggestions. We sincerely apologize for the mistake. To minimize the potential impact of inter-observer variability in sample collection techniques on the positivity rate of avian chlamydia, all samples were primarily collected by veterinarian Zhong, H.T. (Veterinary License Number : A012021450095) and Liang, Y.J. (Veterinary License Number: A012023350270), who followed standardized protocols. Additional personnel were employed in a supportive role to assist with sample acquisition under the supervision of the primary collectors. We have revised the descriptions according to your suggestions and marked them in red on lines 207-211.
Response 2: Thank you for your review and your professional suggestions. According to your suggestions, we have provided the name and location of the animal hospital, as well as details pertaining to the zoo, pet bird market, and duck, pigeon, and chicken farms in Supplementary Materials Table S1. Due to the farms are unwilling to disclose their specific names, we have adopted English abbreviations to represent these entities in this study.
Response 3: Thank you for your review and your professional suggestions. According to your suggestions, we have added the biosafety committee permission number from Guangxi University on line 211, and highlighted it in red.
Response 4: Thank you for your review and your professional suggestions. In this study, the research did not involve animal and/or human subjects and received approval from the biosafety committee of Guangxi University (GXUKE2021-02; March 16, 2021). According to the provisions of Article 7 and Article 8 of the Regulations on Biosafety Management of Pathogenic Microorganism Laboratories and the Catalogue of Pathogen Classification, animal pathogenic microorganisms are classified into distinct categories. Chlamydia is classified as a Class III pathogen, which permits experiments involving pathogen isolation and culture, handling of infectious materials, and diagnostic testing of clinically suspected samples to be conducted in a Biosafety Level 2 (BSL-2) laboratory. All related sample testing procedures were performed in a registered BSL-2 laboratory (Registration Number: Nan Wei Shi Bei Zi [2022] No. 00012), ensuring compliance with the relevant biosafety management requirements.
Comments 8: It would be much easier to follow the “Result” section, if the authors included, in the "Materials and Methods" section, a table showing the number of animals examined from each location—especially the number of species, which only appear later in the tables in the “Results” section.
Response 1: Thank you for your review and your professional suggestions. According to your suggestions, we have provided the detailed information in supplementary materials Table S1. this included the total number of samples, the number of positive samples, locations, name of farms, isolation sources for the zoo, pet bird market, per hospital, and duck, pigeon, goose and chicken farms . Due to the confidentiality concerns, the farms preferred not to disclose their specific names. Thus, we have adopted English abbreviations to represent the company names.
Comments 9: Results section: Much of the data in the Results section is duplicated in the tables, making the section long and confusing. As previously mentioned, there is a lack of statistical support for the findings, making it impossible to draw meaningful conclusions. Also in this section authors discussed an obtained results (line 192-line 199; 221-223) (e.g., lines 192–199 and 221–223). It would be a good idea to add a map showing the sample collection locations.
Response 1: Thank you for your review and your professional suggestions. Following your suggestions, the positive rate of avian chlamydia among birds was analyzed using R software version 4.0.5. The results were denoted as 95% confidence intervals (CI) calculated using the Wilson score interval. Statistical differences among groups of poultry, pet birds, and wild birds were determined through chi-square test. These description have been incorporated on lines 215-218, 223-234, 242-254, 261-270, and highlighted in red. Additionally, Wilson 95% confidence intervals and statistical differences analysis have been added to the results in Table 2, Table 3, and Table 4.
Response 2: Thank you for your review and your professional suggestions. According to your suggestions, we have added the map showing the sample collection locations in Figure 1.
Comments 10: Discussion section: The Discussion is overly long (nearly five pages) and reads more like a review article than a critical analysis of the study's results. There is a lot of repetition from the Results section.
Response 1: Thank you for your review and your professional suggestions. We sincerely agreed with your opinions and have accordingly restructured the discussion section. We have integrated the positivity rates and genotypes of avian chlamydia for each bird species into a single paragraph. This revision aims to minimize redundancy in the results and enhance the discussion of the findings in this study. The revised sections in the discussion are highlighted in red.
Comments 11: I also noticed that there are two “Conclusion” sections—one at the end of the Discussion and one titled “Conclusion.” Please consolidate this.
Response 1: Thank you for your review and your professional suggestions. According to your suggestions, we have have corrected this section, and highlighted on lines 514-520 in red .
Comments 12: In the review dashboard, I did not have access to Table S1.
Response 1: Thank you for pointing this out. We have resubmitted the Table S1 and Table S2.
Comments 13: There are several typos in the bibliography. For example, lines 497, 506, 508, and 511. Chlamydia psittaci and other Latin names should be italicized—please review this section carefully.
Response 1: Thank you for pointing this out. We are terribly sorry for the mistake, and we have corrected the descriptions according to your suggestions in the Reference section and highlighted them with red.
Comments 14: Line 540: it should read “(Psittacula krameri) in France” not “( psittacula krameri ) in france”.
Response 1: Thank you for pointing this out. We are terribly sorry for the mistake, and we have corrected the descriptions according to your suggestions and marked it with red on line 616.
Comments 15: Line 575: “LI, L.; LUTHER, M.; MACKLIN, K.; PUGH, D.; LI, J.; ZHANG, J.; ROBERTS, J.; KALTENBOECK, B.; WANG, C.” – please adjust to the correct reference style.
Response 1: Thank you for pointing this out. We are terribly sorry for the mistake, and we have corrected the descriptions according to your suggestions and marked them with red on line 601.
- Response to Comments on the Quality of English Language
Response 1: We sincerely thank you for highlighting the need for improvement in the English language presentation. We have taken your suggestions and revised the manuscript. These changes included: (1) Correcting grammatical errors and improving sentence structure for better clarity; (2) Ensuring precise and consistent use of scientific terminology; (3) Polishing the overall flow and readability of the text. We hoped these changes could improve the quality of the manuscript.
Reviewer 2 Report
Comments and Suggestions for Authors
The manuscript “Prevalence and genetic characteristics of avian chlamydia in birds in Guangxi province, southwestern China” by Jian ming Long and coauthors conducted an epidemiological study of Chlamydia in various bird species. The authors performed PCR detection and sequencing for species and genotype characterization. They detected a significant prevalence of C. psittaci and C. gallinacea. The results are interesting in terms of monitoring. I only have a few comments that I think could be improved.
• Table 1: In the Genotype column: correct "D. gallinacea." In the host column: standardize all names to either the common name or the scientific name. Scientific names should always appear in italics.
• Section 2.3: Indicate the size of the amplified fragment for sequencing (line 130). In addition, indicate the method used to perform the sequence alignment and the nucleotide substitution model used to construct the phylogenetic tree.
• Table 2: Standardize the definitions of duck and pigeon farms with respect to lines 151-153.
• Table 3: The positivity values do not correspond to those indicated in lines 158-159. Similarly, the scientific name for zebra finch does not correspond to those indicated in line 162.
• Lines 36 and 62: It should be “range”.
• Check the references. Documents #16, 30, and 39 are the same.
• Verify that scientific names are in italics throughout the manuscript. i.e., lines 250, 285.
• Line 282: Should be ref 53?
• References 55 and 56 are not included in the text. Verify all the references.
Author Response
Response to Reviewer 2 Comments
- Summary
We thank the reviewer for the kind consideration and constructive comments on our manuscript. We have carefully revised the manuscript and provided a detailed point-by-point response, with changes highlighted in red. We hope these changes will enhance the quality of our manuscript.
- Questions for General Evaluation
Does the introduction provide sufficient background and include all relevant references?/ Yes
Response: Thank you for your review and your professional suggestions.
Is the research design appropriate? /Yes
Response: Thank you for your review and your professional suggestions.
Are the methods adequately described?/Can be improved
Response: Thank you for your review and your professional suggestions. We are terribly sorry to make this mistake. A 1108-bp segment of the ompA gene, which includes variable domains (VDs) I-IV, amplified using nested PCR. And we added this description on line 179, and highlighted in red. Additionally, we have added the description according to your suggestions on lines 195-198, and highlighted in red.
Are the results clearly presented? /Can be improved
Response: Thank you for your review and your professional suggestions. We are terribly sorry to make this mistake. We have corrected the description on lines 228-232, and highlighted in red.
Are the conclusions supported by the results? /Yes
Response: Thank you for your review and your professional suggestions.
Are all the figures and tables clear and well-presented? /Can be improved
Response: Thank you for your review and your professional suggestions. We sincerely apologize for the mistake. We have corrected this mistake according to your suggestions in Table S2, and ensure consistency in the host name in Table 1, Table S1 and Table S2, .
- Point-by-point response to Comments and Suggestions for Authors
Comment 1: Table 1: In the Genotype column: correct "D. gallinacea." In the host column: standardize all names to either the common name or the scientific name. Scientific names should always appear in italics.
Response 1: Thank you for your review and your professional suggestions. We are terribly sorry to make this mistake. We have corrected this mistake according to your suggestions in Table S2, and ensure consistency in the host name in Table 1, Table S1 and Table S2, .
Comment 2: Section 2.3: Indicate the size of the amplified fragment for sequencing (line 130). In addition, indicate the method used to perform the sequence alignment and the nucleotide substitution model used to construct the phylogenetic tree.
Response 1: Thank you for your review and your professional suggestions. We are terribly sorry to make this mistake. A 1108-bp segment of the ompA gene, which includes variable domains (VDs) I-IV, amplified using nested PCR. And we added this description on line 179, and highlighted in red.
Response 2: Thank you for your review and your professional suggestions. We have added this description according to your suggestions on lines 195-198, and highlighted in red.
Comment 3: Table 2: Standardize the definitions of duck and pigeon farms with respect to lines 151-153.
Response 1: Thank you for your review and your professional suggestions. We are terribly sorry to make this mistake. We have corrected this description on lines 228-232, and highlighted in red.
Comment 4: Table 3: The positivity values do not correspond to those indicated in lines 158-159. Similarly, the scientific name for zebra finch does not correspond to those indicated in line 162.
Response 1: Thank you for your review and your professional suggestions. We are terribly sorry to make this mistake. We have corrected this description on lines 243-244, and highlighted in red. And we have ensured consistency in the host name in Table 3.
• Lines 36 and 62: It should be “range”.
Response 2: Thank you for your review and your professional suggestions. We are terribly sorry to make this mistake. We have corrected this description on line 39 and 63, and highlighted in red.
• Check the references. Documents #16, 30, and 39 are the same.
Response 3: Thank you for your review and your professional suggestions. We are terribly sorry to make this mistake. We have corrected this description on line 578, and highlighted in red.
• Verify that scientific names are in italics throughout the manuscript. i.e., lines 250, 285.
Response 4: Thank you for your review and your professional suggestions. We are terribly sorry to make this mistake. We have corrected this description on lines 343 and 419, and highlighted in red.
• Line 282: Should be ref 53?
• References 55 and 56 are not included in the text. Verify all the references.
Response 5: Thank you for your review and your professional suggestions. We are terribly sorry to make this mistake. We have corrected this mistake in References section, and highlighted in red on lines 394 and 397.
Reviewer 3 Report
Comments and Suggestions for Authors
I have carefully reviewed the manuscript entitled "Prevalence and genetic characteristics of avian chlamydia in birds in Guangxi province, southwestern China". The topic is relevant and timely, addressing the prevalence and genetic diversity of Chlamydia spp. in different avian hosts. The findings have potential implications for animal and public health, particularly given the zoonotic potential of Chlamydia psittaci and the emerging role of other Chlamydia species. The manuscript is generally well written and presents a considerable amount of field and molecular data. However, several substantial issues should be addressed before the manuscript can be considered for publication.
Major Comments
- Experimental design and sampling strategy
The description of the study design is insufficient. The authors must clearly explain: - How farms, households, or locations were selected.
- The criteria for sample size determination for each bird species / production system.
- Whether repeated sampling was conducted in the same units over the study period.
- The temporal distribution of sampling (season, year), and how temporal variation was handled in the analyses.
- Reference strains selection
The criteria for selecting reference strains for phylogenetic analyses are not described. Please provide a clear justification for their inclusion, indicating their geographic origin, host species, and relevance to the study. - Prevalence versus positivity
If the study design does not ensure representativeness, the authors must avoid using “prevalence” and instead report “positivity rates.” This should be corrected consistently in the abstract, results, discussion, and conclusions. - Confidence intervals
For all proportions reported (positivity or prevalence), 95% confidence intervals should be calculated and included in the text, tables, and figures. This will strengthen the statistical interpretation of the findings. - Selection of samples for phylogenetic analysis
The manuscript states that 74 samples were used for phylogenetic analysis. Please describe the selection criteria for these samples, including whether they represent all positive hosts proportionally, or if selection was based on Ct values, geographic distribution, or species diversity. - Methodology for potential bird-to-human transmission risk
The manuscript highlights the zoonotic potential of Chlamydia spp., but does not clearly describe any methodology or criteria used to assess the potential risk of bird-to-human transmission. If such an evaluation was performed, the methods and indicators used should be explicitly described. If not, this should be clarified and discussed as a limitation. - Discussion update
The discussion section must be revised to reflect any changes made in the Materials and Methods and Results sections after addressing the above points. Comparisons with recent literature should be strengthened, particularly regarding zoonotic implications, diversity of Chlamydia spp., and possible cross-species transmission pathways.
Minor Comments
- Relevance of the study location
The introduction should include a short justification of why Guangxi province is an important region for this type of study (e.g., poultry production scale, avian biodiversity, role in national or international trade, border proximity). - Abstract
- Include 95% confidence intervals for all reported prevalence/positivity values.
- Ensure consistency in terminology (positivity vs. prevalence).
Author Response
Response to Reviewer 3 Comments
- Summary
We thank the reviewer for the kind consideration and constructive comments on our manuscript. We have carefully revised the manuscript and provided a detailed point-by-point response, with changes highlighted in red. We hope these changes will enhance the quality of our manuscript.
- Questions for General Evaluation
Does the introduction provide sufficient background and include all relevant references?/ Yes
Response: Thank you for your review and your professional suggestions.
Is the research design appropriate? /Must be improved
Response: Thank you for your review and your professional suggestions. We have added the description and table according to your suggestions in Materials and Methods section in line 100-158 and line 205-218, and highlighted these changes in red.
Are the methods adequately described?/Must be improved
Response: Thank you for your review and your professional suggestions. We have added the description and table according to your suggestions in Materials and Methods section on lines 100-158 and 205-218, and highlighted these changes in red.
Are the results clearly presented? /Can be improved
Response: Thank you for your review and your professional suggestions. We have added the description and calculation (including 95% confidence intervals and Statistical differences) according to your suggestions in Result section on lines 223-234, 242-254, and 261-270, and highlighted these changes in red.
Are the conclusions supported by the results? /Can be improved
Response: Thank you for your review and your professional suggestions. We have added the description and calculation (including 95% confidence intervals and Statistical differences) according to your suggestions in Result section on lines 223-234, 242-254, and 261-270, and highlighted these changes in red.
Are all the figures and tables clear and well-presented? /Can be improved
Response: Thank you for your review and your professional suggestions. We have added the tables and figure according to your suggestions in Materials and Methods, Result, and Supplementary Materials section. Table1 found on line 205, while Table S1 and Table S2 have also been added. Figure 1, which denotes the sampling area presented on line 121.
- Point-by-point response to Comments and Suggestions for Authors
Comment 1: Experimental design and sampling strategy: The description of the study design is insufficient. The authors must clearly explain:
- How farms, households, or locations were selected.
- The criteria for sample size determination for each bird species / production system.
- Whether repeated sampling was conducted in the same units over the study period.
- The temporal distribution of sampling (season, year), and how temporal variation was handled in the analyses.
Response 1: Thank you for your review and your professional suggestions. We have added the description and tables according to your suggestions in Materials and Methods section on lines 100-158, and highlighted these changes in red. According to your suggestions, we have provided the detailed information including the total number of samples, the number of positive samples, location, name of farm, isolation source for the zoo, pet bird market, per hospital, and duck, pigeon, goose and chicken farms in Supplementary Materials Table S1.
Response 2: Thank you for your review and your professional suggestions. We have added the description and table according to your suggestions in Materials and Methods section on lines 100-158, with marking them in red. According to your suggestions, we have provided the detailed information including the total number of samples, the number of positive samples, location, name of farm, isolation source for the zoo, pet bird market, per hospital, and duck, pigeon, goose and chicken farms in Supplementary Materials Table S1.
Response 3: Thank you for pointing this out. We fully agree with your suggestions. We didn’t repeat sampling in the same unites over the study period except the zoo. The reasons are as follows: (1) The zoo implemented an annual vaccination program for pet birds exhibited to the public during the spring and autumn seasons. We could only collected samples from pet birds during their vaccination. (2) The collection of wild bird samples were predominantly relied on veterinary submissions, community reports, or wildlife rehabilitation centers. Thus, the sample source is limited, making it difficult to repeat sampling. (3) Samples from poultry were challenging to repeat sampling due to their dispersed geographic locations, extensive sampling range, and the large volume of samples required. Nevertheless, we sincerely agree with your suggestion and will incorporate this factor into the ongoing monitoring of avian chlamydia in birds, thereby refining the epidemiological data on this pathogen.
Response 4: Thank you for pointing this out. We acknowledge that our study did not incorporate temporal factors (e.g., seasonality, year-to-year variation) in its sampling design or statistical analysis. This limitation stemmed from the cross-sectional nature of our study, which was primarily focused on the baseline positivity rate of avian chlamydia infection. Additionally, the sampling process did not include detailed temporal metadata (e.g., seasonal categorizations, year) to enable such analyses. We fully agree that temporal variation is critical for understanding the epidemiology of avian chlamydia and would like to emphasize its significance in influencing the prevalence of avian chlamydia in future research endeavors. To address this limitation, we plan to design longitudinal studies to monitor seasonal and interannual patterns of infection. Additionally, we will ensure the collection of standardized temporal metadata (e.g., sampling dates, climatic conditions) in future studies. We also plan to collaborate with long-term poultry monitoring programs (such as those for avian influenza) to integrate time-series data for more robust analyses. We sincerely appreciate your suggestion, as it has highlighted a key direction for improving our future research on avian chlamydiosis.
Comment 2: Reference strains selection
The criteria for selecting reference strains for phylogenetic analyses are not described. Please provide a clear justification for their inclusion, indicating their geographic origin, host species, and relevance to the study.
Response 1: Thank you for your review and your professional suggestions. According to your suggestions, we have added the detailed information including geographic origin, host species in supplementary materials Table S2. Additionally, We have added the description according to your suggestions in Materials and Methods section on line 200-205, and highlighted in red.
Comment 3: Prevalence versus positivity
If the study design does not ensure representativeness, the authors must avoid using “prevalence” and instead report “positivity rates.” This should be corrected consistently in the abstract, results, discussion, and conclusions.
Response 1: Thank you for your review and your professional suggestions. We sincerely agree with your suggestions. According to your suggestions, we have corrected the relative descriptions throughout the manuscript and marked these changes in red.
Comment 4: Confidence intervals
For all proportions reported (positivity or prevalence), 95% confidence intervals should be calculated and included in the text, tables, and figures. This will strengthen the statistical interpretation of the findings.
Response 1: Thank you for your review and your professional suggestions. According to your suggestions, the positivity rate of avian chlamydia among birds was analyzed using R software version 4.0.5. The results were denoted as 95% confidence intervals (CI) calculated using the Wilson score interval. Statistical differences among groups of poultry, pet birds, and wild birds were determined through chi-square test. These description have been added on lines 211-215, 223-234, 242-254, 261-270, 327-330, 337-343, 367-369, 377-379, 398-399, 423-426, 443-445, 453-455, and 475-478, and were highlighted in red. Additionally, Table 2, Table 3, and Table 4 have been updated to include Wilson 95% confidence intervals and statistical differences analysis.
Comment 5: Selection of samples for phylogenetic analysis
The manuscript states that 74 samples were used for phylogenetic analysis. Please describe the selection criteria for these samples, including whether they represent all positive hosts proportionally, or if selection was based on Ct values, geographic distribution, or species diversity.
Response 1: Thank you for your review and your professional suggestions. According to your suggestions, we have added the descriptions in lines 175-181 and 191-193, and highlighted in red. Additionally, the detailed information including source, host, location and accession numbers were listed in Table 1.
Response 2: Thank you for your review and your professional suggestions. The nucleotide sequences of the ompA gene used in this study represented all positive samples from poultry, as well as majority of those from pet birds and wild birds. The low-quality sequences were filtered out, and the remaining 74 representative sequences were selected for phylogenetic tree construction.
Comment 6: Methodology for potential bird-to-human transmission risk
The manuscript highlights the zoonotic potential of Chlamydia spp., but does not clearly describe any methodology or criteria used to assess the potential risk of bird-to-human transmission. If such an evaluation was performed, the methods and indicators used should be explicitly described. If not, this should be clarified and discussed as a limitation.
Response 1: Thank you for your review and your professional suggestions. We sincerely apologize for not implementing a methodology to assess the potential risk of bird-to-human transmission. And we have added the limitation in the discussion section on lines 411-416 and lines 508-509, and highlighted in red.
Comment 7: Discussion update
The discussion section must be revised to reflect any changes made in the Materials and Methods and Results sections after addressing the above points. Comparisons with recent literature should be strengthened, particularly regarding zoonotic implications, diversity of Chlamydia spp., and possible cross-species transmission pathways.
Response 1: Thank you for your review and your professional suggestions. We have revised this section according to your suggestions on lines 327-329, 337-342, 367-368, 376-378, 398-399, 422-423, 442-445, 452-454, and 476-477, and highlighted in red.
Response 2: Thank you for your review and your professional suggestions. We have added the recent literature to the discussion section according to your suggestions on lines 333-334, 361-363, 368-369, 405-407, 422-426, 439-441, 444-446, 469-470, and 490-491, and highlighted in red.
Reference:
- De Meyst, A.; De Clercq, P.; Porrez, J.; Geens, T.; Braeckman, L.; Ouburg, S.; Morré, S.A.; Vanrompay, D. Belgian cross-sectional epidemiological study on zoonotic avian chlamydia in chickens. Microorganisms. 2024, 12, 193.
- Solorzano-Morales, A.; Dolz, G. Molecular characterization of Chlamydia species in commercial and backyard poultry farms in Costa Rica. Infect.2022, 150.
- Hou, L.; Jia, J.; Qin, X.; Fang, M.; Liang, S.; Deng, J.; Pan, B.; Zhang, X.; Wang, B.; Mao, C.; et al. Prevalence and genotypes of Chlamydia psittaci in birds and related workers in three cities of China. PLoS One. 2024, 19, e308532.
- Liu, S.; Li, K.; Hsieh, M.; Chang, P.; Shien, J.; Ou, S. Prevalence and genotyping of Chlamydia psittaci from domestic waterfowl, companion birds, and wild birds in Taiwan. Vector-Borne Zoonotic Dis.2019, 19, 666-673.
- Le Gall-Ladevèze, C.; Vollot, B.; Hirschinger, J.; Lèbre, L.; Aaziz, R.; Laroucau, K.; Guérin, J.; Paul, M.; Cappelle, J.; Le Loc H, G. Limited transmission of avian influenza viruses, avulaviruses, coronaviruses and Chlamydia at the interface between wild birds and a free-range duck farm. Vet. Res.2025, 56, 36.
- Soon, X.Q.; Gartrell, B.; Gedye, K. Presence and shedding of Chlamydia psittaci in waterfowl in a rehabilitation facility and in the wild in New Zealand. Z. Vet. J. 2021, 69, 240-246.
- Qin, X.; Huang, J.; Yang, Z.; Sun, X.; Wang, W.; Gong, E.; Cao, Z.; Lin, J.; Qiu, Y.; Wen, B.; et al. Severe community-acquired pneumonia caused by Chlamydia psittaci genotype E/B strain circulating among geese in Lishui city, Zhejiang province, China. Microbes Infect. 2022, 11, 2715-2723.
- Zhang, Z.; Zhou, H.; Cao, H.; Ji, J.; Zhang, R.; Li, W.; Guo, H.; Chen, L.; Ma, C.; Cui, M.; et al. Human-to-human transmission of Chlamydia psittaci in China, 2020: an epidemiological and aetiological investigation. The Lancet. Microbe. 2022, 3, e512-e520.
- Zhang, R.; Fu, H.; Luo, C.; Huang, Z.; Pei, R.; Di, Y.; Zhu, C.; Peng, J.; Hu, H.; Chen, S.; et al. Chlamydia psittaci detected at a live poultry wholesale market in central China. BMC Infect. Dis.2024, 24, 585.
- AL-sultani, S. Molecular detection and genotyping of Chlamydia psittaci in domestic pigeons and human contacts in Baghdad city. Kerbala Journal of Veterinary Medical Sciences. 2025, 1.
- Ko, J.C.K.; Choi, Y.W.Y.; Poon, E.S.K.; Wyre, N.; Go, J.L.L.; Poon, L.L.M.; Sin, S.Y.W. Prevalence and genotypes of Chlamydia psittaci in pet birds of Hong Kong. PLoS One. 2024, 19, e306528.
- Lee, H.J.; Lee, O.M.; Kang, S.I.; Yeo, Y.G.; Jeong, J.Y.; Kwon, Y.K.; Kang, M.S. Prevalence of asymptomatic infections of Chlamydia psittaci in psittacine birds in Korea. Zoonoses Public Health. 2023, 70, 451-458.
- Kasimov, V.; Dong, Y.; Shao, R.; Brunton, A.; Anstey, S.I.; Hall, C.; Chalmers, G.; Conroy, G.; Booth, R.; Timms, P.; et al. Emerging and well-characterized chlamydial infections detected in a wide range of wild Australian birds. Emerg. Dis.2022, 69.
Comment 8: Relevance of the study location
The introduction should include a short justification of why Guangxi province is an important region for this type of study (e.g., poultry production scale, avian biodiversity, role in national or international trade, border proximity).
Response 1: Thank you for your review and your professional suggestions. We have added the description to according to your suggestions on lines 88-92, and highlighted in red.
Comment 9: Abstract
- Include 95% confidence intervals for all reported prevalence/positivity values.
- Ensure consistency in terminology (positivity vs. prevalence).
Response 1: Thank you for your review and your professional suggestions. We have added the description to according to your suggestions in lines 23-30, and highlighted in red.
Response 2: Thank you for your review and your professional suggestions. We fully agree with your suggestions. According to your suggestions, we have corrected the relative descriptions throughout the manuscript and and highlighted these changes in red.
- Response to Comments on the Quality of English Language
Response 1: We sincerely thank you for highlighting the need for improvement in the English language presentation. We have taken your suggestions and revised the manuscript. These changes included: (1) Correcting grammatical errors and improving sentence structure for better clarity; (2) Ensuring precise and consistent use of scientific terminology; (3) Polishing the overall flow and readability of the text. We hoped these changes could improve the quality of the manuscript.
Reviewer 4 Report
Comments and Suggestions for Authors
Long, Zhong, Deng, Yang, Chen, Liang, Chen, Liang, Wei, Wei, & Huang: Prevalence and Genetic Characteristics of Avian Chlamydia in Birds in Guangxi Province, Southwestern China
The authors investigate the prevalence and genetic diversity of avian Chlamydia spp. in domestic poultry, pet birds, and wild birds in Guangxi Province, China. Using PCR screening of 1,744 samples and phylogenetic analysis of the ompA gene, they report an overall prevalence of 28.21%, identify diverse C. psittaci genotypes and C. gallinacea, and note new host–genotype associations. The findings highlight zoonotic potential and call for ongoing surveillance. The paper is generally very well written, with theory and results that are easy to follow.
The paper presents original and novel data, such as the first detection of C. gallinacea in pet birds and C. psittaci genotype B in ducks in China, and provides much-needed data from a geographic region with limited epidemiological information. I especially wish to emphasize the large and diverse sample set from poultry farms, pet markets, zoos, and wild birds obtained by the authors. The inclusion of multiple avian orders further improves understanding of the host range. The methods are valid and appropriate. I found it important that the discussion clearly connects findings to zoonotic risk, including links to genotypes involved in human psittacosis cases. Overall, this submission presents novel, well-supported findings with significant relevance to avian and public health. The data fill an important knowledge gap, though improvements in presentation, species identification, and genomic resolution would strengthen the impact.
Some weaknesses of this study have already been mentioned by the authors in the Discussion, and I have only a few additional suggestions to slightly improve the text:
(1) Minor grammatical inconsistencies and occasional awkward phrasing reduce clarity in places; however, I do not believe the language requires major improvements. If the journal insists, please check the style and grammar once more.
(2) I suggest improving species reporting: use consistent scientific names for all species, and specify subspecies where relevant to epidemiology. For example, in Table 1, some species are listed by their scientific names while others are given in English. Please ensure consistency.
(3) If possible, better integrate human epidemiology by including, where available, surveillance or case data from local human populations to directly link avian findings to zoonotic outcomes.
Specific points:
Line 48 and throughout: “most infected birds[1-3]”: insert a space between “birds” and “[1-3]”.
Lines 139, 201, 454: ompA; line 285: C. psittaci. These terms must be italicized.
Author Response
We sincerely apologize for inadvertently submitting an incorrect response to the comments raised by the fourth reviewer.
Response to Reviewer 4 Comments
- Summary
We thank the reviewer for the kind consideration and constructive comments on our manuscript. We have carefully revised the manuscript and provided a detailed point-by-point response, with changes highlighted in red. We hope these changes will enhance the quality of our manuscript.
- Questions for General Evaluation
Response: Thank you for your review and your professional suggestions.
Is the research design appropriate? /Yes
Response: Thank you for your review and your professional suggestions.
Are the methods adequately described?/Yes
Response: Thank you for your review and your professional suggestions.
Are the results clearly presented? /Yes
Response: Thank you for your review and your professional suggestions.
Are the conclusions supported by the results? /Yes
Response: Thank you for your review and your professional suggestions.
Are all the figures and tables clear and well-presented? /Yes
Response: Thank you for your review and your professional suggestions.
- Point-by-point response to Comments and Suggestions for Authors
Comment 1: Minor grammatical inconsistencies and occasional awkward phrasing reduce clarity in places; however, I do not believe the language requires major improvements. If the journal insists, please check the style and grammar once more.
Response 1: We sincerely thank you for highlighting the need for improvement in the English language presentation. We have taken your suggestions and revised the manuscript. These changes included: (1) Correcting grammatical errors and improving sentence structure for better clarity; (2) Ensuring precise and consistent use of scientific terminology; (3) Polishing the overall flow and readability of the text. We hoped these changes could improve the quality of the manuscript.
Comment 2: I suggest improving species reporting: use consistent scientific names for all species, and specify subspecies where relevant to epidemiology. For example, in Table 1, some species are listed by their scientific names while others are given in English. Please ensure consistency.
Response 1: Thank you for your review and your professional suggestions. We sincerely apologize for the mistake. We have revised the species name for consistency in English name according to your suggestions in Table 1, Table S1 and Table S2. Also, we have specified subspecies relevant to epidemiology on lines 244-255 and 263-268, 301, and 313, and highlighted these changes in red.
Comment 3: If possible, better integrate human epidemiology by including, where available, surveillance or case data from local human populations to directly link avian findings to zoonotic outcomes.
Response 1: Thank you for your review and your professional suggestions. We sincerely apologize for that no swab samples were collected from human due to personal preferences and ethical considerations, which limited our ability to assess of the zoonotic potential of avian chlamydia. This limitation has been addressed in the discussion section on lines 411-414 and 508-509, with changes marked in red. Meanwhile, We fully agree with your suggestions and will continue surveillance ofavian chlamydia in both birds and humans to refine the epidemiological data on this pathogen.
Comment 4: Line 48 and throughout: “most infected birds[1-3]”: insert a space between “birds” and “[1-3]”.
Response 1: Thank you for your review and your professional suggestions. We are terribly sorry to make this mistakes. We have corrected this mistakes throughout the manuscript according to your suggestions and marked these changes in red.
Comment 5: Lines 139, 201, 454: ompA; line 285: C. psittaci. These terms must be italicized.
Response 1: Thank you for your review and your professional suggestions. We are terribly sorry to make these mistakes. We have corrected these mistakes according to your suggestions and marked them in red on lines 200, 205, 419 and 509.